# LEARNING CONTINUOUS ENVIRONMENT FIELDS VIA IMPLICIT FUNCTIONS

Xueting Li[1], Shalini De Mello[2], Xiaolong Wang[3], Ming-Hsuan Yang[1], Jan Kautz[2], and Sifei Liu[2]

[1]UC Merced
[2]NVIDIA
[3]UC San Diego

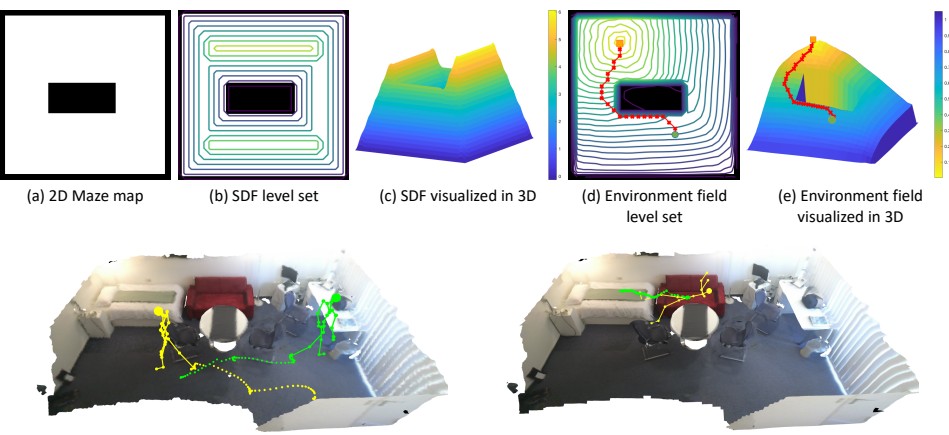

(a) 2D Maze map    (b) SDF level set    (c) SDF visualized in 3D    (d) Environment field level set    (e) Environment field visualized in 3D

(f) Searched trajectories in 3D indoor environments

Figure 1: **Implicit environment fields in 2D and 3D space.** With implicit functions, we learn a continuous environment field that encodes the reaching distance between any pair of points in 2D and 3D space. We present utilization of the environment field for 2D maze navigation in (d)-(e) and human scene interaction modeling in 3D scenes in (f). Note that we flip the environment field in (e) upside down for visualization purposes.

## ABSTRACT

We propose a novel scene representation that encodes *reaching distance* – the distance between any position in the scene to a goal along a feasible trajectory. We demonstrate that this environment field representation can directly guide the dynamic behaviors of agents in 2D mazes or 3D indoor scenes. Our environment field is a continuous representation and learned via a neural implicit function using discretely sampled training data. We showcase its application for agent navigation in 2D mazes, and human trajectory prediction in 3D indoor environments. To produce physically plausible and natural trajectories for humans, we additionally learn a generative model that predicts regions where humans commonly appear, and enforce the environment field to be defined within such regions. Extensive experiments demonstrate that the proposed method can generate both feasible and plausible trajectories efficiently and accurately.

## 1 INTRODUCTION

Scene understanding aims to analyze and interpret a given environment. The past few years have witnessed tremendous success in scene representation learning for semantic segmentation (Long et al., 2015; Li et al., 2017), 3D scene reconstruction (Sitzmann et al., 2020; 2019), and depth estimation (Godard et al., 2019). The learned representations can be used as inputs for training embodied tasks including visual navigation (Sax et al., 2018; Zhou et al., 2019) and robotic manipulation (Chen et al., 2020; Yen-Chen et al., 2020). This overall process falls into a two-step paradigm, which first learns a *static* scene representation, and then trains an agent to interact with the scene. While this progress is exciting, we ask for exploring an alternative research direction: Can the scene

representation itself be *dynamically interactive*? That is, instead of training a Reinforcement Learning agent on top of the scene, we encode the *dynamic* functioning (i.e., the ability to control and guide the agent's interactions) in the scene representation in one-step training.

Specifically, we study a representation that guides an embodied agent to navigate inside the scene. We propose a scene representation dubbed as the *environment field* that encodes the *reaching distance* – distance from any position to a given goal location along a feasible and plausible trajectory. Such an environment field needs to be continuous and densely covering the entire scene. With this continuous scene representation, given an agent in any locations in the scene, we can perform iterative optimization (e.g., gradient descent) over the continuous field to move the agent to a goal. The optimization objective is to minimize the reaching distance, and the agent reaches a given goal when the reaching distance equals to zero.

Compared to search-based algorithms (Sethian, 1996; LaValle, 1998; Kavraki et al., 1996), the proposed environment field enjoys two key advantages: a) *Continuity*. Search-based methods (Sethian, 1996) assume a discrete searching space, while the proposed environment field is continuous and can easily generalize to continuous scenes such as a 3D indoor environment. b) *Efficiency*. Instead of applying the time-consuming searching algorithm to every new environment or goal position, a single neural network is learned to efficiently produce different environment fields for different scenes or goal positions, with a forward pass.

How to learn this continuous environment field? Naively learning such a dense field in a supervised matter requires ground truth reaching distance annotation at all possible locations in the scene, which are infinitely many and thus impossible to collect. In this paper, we leverage neural implicit functions (Park et al., 2019; Xu et al., 2019; Sitzmann et al., 2020; Xu et al., 2019; Mescheder et al., 2019) that can learn a continuous field using discretely sampled training data. Specifically, we design an implicit neural network that takes the coordinates of a position in the scene as input and outputs its reaching distance to a given goal location. Even though this implicit neural network is trained using discretely sampled position and reaching distance pairs, yet it produces a dense environment field of a scene. During inference, the reaching distance between *any* position in the scene and the goal can be efficiently queried via a fast network forward pass and used to navigate an agent towards the goal.

We showcase the application of the proposed environment field in both human trajectory modeling in 3D indoor environments as well as agent navigation in 2D mazes and observe comparable if not better results than state-of-the-art methods. Specially, when applying the environment field to human trajectory modeling, we additionally propose and learn an *accessible region* – plausible regions for humans to appear in the scene via a variational auto-encoder (VAE). This accessible region can be implicitly encoded into the environment field so that it can guide humans to avoid locations that do not conform to common human behaviors (e.g., navigate a human to travel around rather than under a desk).

The main contributions of this work are:

- We propose the environment field – a novel scene representation that facilitates direct guidance of dynamic agent behaviors in a scene.
- We learn dense environment fields via implicit functions using discretely sampled training pairs.
- We apply the environment fields to human trajectory prediction in 3D indoor environments and agent navigation in 2D mazes, where both feasible and plausible trajectories can be produced efficiently and accurately.

## 2  RELATED WORK

**Scene representations.**  Scene understanding aims at parsing and interpreting an environment. Prior approaches learn various scene representations comprising of semantic segmentation (Long et al., 2015; Li et al., 2017), depth (Godard et al., 2019) and meshes (Sitzmann et al., 2020). Nevertheless, these representations only capture static scene attributes while the proposed environment field aims to encode the dynamic functioning of a scene.

**Path planning.**  Classical path planning algorithms including Breath First Searching (BFS), Dijkstra's algorithm (Dijkstra, 1959) and the Fast Marching Method (FMM) (Sethian, 1996; Valero-Gomez et al., 2015) require applying the searching algorithm to every new environment or goal position, and hence are not suitable for dynamically changing environments. On the contrary, our

environment field generalizes to changing goals and environments during inference and is thus computationally efficient. Instead of solving path planning numerically, sampling-based methods such as Rapidly Exploring Random Tree (RRT) (LaValle, 1998; LaValle et al., 2001; Bry & Roy, 2011) or Probabilistic Roadmaps (PRM) (Kavraki et al., 1996; 1998) grow and search a tree by sampling feasible locations in the scene. However, the efficiency and performance of such methods depend on the sampling density and algorithm. In contrast, we learn a continuous environment field that predicts the next optimal move instantly for any location in the scene. The potential field algorithm manually computes and combines an attractive and repulsive field and is prone to local minimas, while our algorithm learns a single field in a data-driven way.

The proposed environment field can also be considered as a value function used in learning-based path planning methods (Tamar et al., 2016; Campero et al., 2020; Al-Shedivat et al., 2018; Chaplot et al., 2021). The reaching distance at each location is essentially the reward the agent gets while trying to reach the goal. However, compared to other learning-based path planning algorithms (Tamar et al., 2016; Campero et al., 2020; Al-Shedivat et al., 2018), the environment field enjoys two major benefits. First, it produces the environment field for all locations in the scene with a single network forward pass (in discrete cases such as 2D mazes), instead of predicting optimal steps sequentially, which requires multiple network forward passes. Second, it utilizes implicit functions instead of convolution neural networks and is thus more flexible to generalize to continuous scenarios such as human navigation in 3D scenes.

**Implicit neural representations.** Implicit neural representations have recently attracted much attention in computer vision due to their continuity, memory-efficiency, and capacity to capture complex object details. Numerous implicit neural representations have been proposed to reconstruct 3D objects (Park et al., 2019; Xu et al., 2019; Michalkiewicz et al., 2019; Tulsiani et al., 2020; Mescheder et al., 2019), scenes (Sitzmann et al., 2019; Jiang et al., 2020; Peng et al., 2020; Mildenhall et al., 2020), humans (Atzmon & Lipman, 2020; Saito et al., 2019; 2020) or even audio and video signals (Sitzmann et al., 2020). Instead of reconstruction, in this work, we use an implicit neural function to encode a continuous environment field and target at agent navigation. A more detailed overview of implicit neural networks can be found in Section 3.1.

**Affordance prediction.** Affordance prediction models the interactions between humans and objects or scenes (Koppula & Saxena, 2014; Chuang et al., 2018; Zhu et al., 2014; Koppula et al., 2013; Li et al., 2019; Karunratanakul et al., 2020). One line of work describes static human-scene interactions by generating human poses (Wang et al., 2017; Li et al., 2019) or human body meshes (Zhang et al., 2020b;a) given scene images or point clouds. Another line of work (Cao et al., 2020; Monszpart et al., 2019) models dynamic human-scene interactions by predicting a sequence of future human poses given pose history. The proposed method falls into the second category, where we predict plausible human motion sequences containing different actions (e.g., walking and sitting down) while avoiding collisions in 3D scenes. The proposed approach enjoys two major advantages compared to existing work (Cao et al., 2020; Monszpart et al., 2019): a) Our work does not require any past human motion history as a reference. Once the environment field of a scene is encoded by an implicit function, an agent can be navigated from any position to the goal position. b) Our work can generate much longer navigation paths than existing work thanks to the learned environment field that covers the entire scene.

## 3 ENVIRONMENT FIELD

We first introduce the environment field using agent navigation in 2D mazes as a toy example. A real-world application in human trajectory modeling is discussed in Section 4.

### 3.1 OVERVIEW OF IMPLICIT NEURAL NETWORKS

Implicit neural networks have been widely used to represent a 3D object or scene as a signed distance function (SDF). The input to an implicit neural network is the coordinates of a 3D point ($x \in \mathrm{R}^3$) and the output is the signed distance ($s \in R$) from this point to the nearest object surface: $s = SDF(x)$, where $s > 0$ point is inside the object surface and $s < 0$ when the point is outside the object surface. A desirable property of an implicit neural network is that it learns a continuous surface of an object given discretely sampled training pairs $X := (x, s)$, i.e., it is able to predict the SDF value for points not in the training set $X$ once trained. During inference, a continuous object surface can be explicitly recovered by extracting an arbitrarily number of points with zero SDF values, i.e., points on the object surface (Lorensen & Cline, 1987).

## 3.2 Environment Field Learning via Neural Implicit Networks

In this work, we take advantage of the continuity property of implicit neural networks described above and learn a continuous environment field from discretely sampled location and reaching distance pairs. Specifically, we represent the environment field as a mapping from location coordinates $x \in \mathrm{R}^2$ to its reaching distance $u(x) \in \mathrm{R}$ towards a given goal and learn an implicit neural network $f_\theta$ (see Fig. 3 (a)) such that:

$$\hat{u}(x) = f_\theta(x). \tag{1}$$

To collect a set of training pairs $X := \{x, u(x)\}$, we resort to an analytical method, e.g., fast marching method (FMM) (Sethian, 1996), that solves $u(x)$ for each cell $x$ in a discrete grid. We then train the implicit function $f_\theta$ to regress the reaching distance $u(x)$ computed by the FMM for each cell $x \in X$ using the $L_1$ loss function:

$$\mathcal{L}(f_\theta(x), u(x)) = |f_\theta(x) - u(x)|. \tag{2}$$

Thanks to the continuity of implicit functions, the learned implicit neural network is able to predict the reaching distance even for points outside the training set $X$, enabling us to query the reaching distance for any location in the scene during inference.

## 3.3 Trajectory Search using Environment Field

Given the learned implicit function in Section 3.2, we now describe how to search a trajectory from a starting position $x_s$ to the goal position $x_e$. Since the learned implicit neural network is continuous, one intuitive solution for navigation is through a gradient descent method (see Fig. 2(c)). We first feed the current location $x_c$ of an agent into the learned implicit function. Then, by minimizing the output reaching distance, the agent can be nudged by the gradients $\Delta x_c$ back-propagated from the implicit function to a new position $x_c - \Delta x_c$, where the reaching distance is smaller, i.e., closer to the goal. Starting at $x_s$, we repeat the above pro-

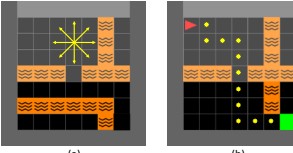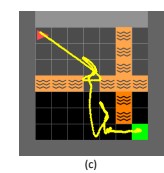

Figure 2: **Trajectory searching.** (a) The 8 possible directions that an agent can take in a 2D maze. (b) Searched trajectory by querying the environment field values of positions the agent can reach within one step. (c) Searched trajectory by gradient descent. The red triangle and the green square represent the starting and goal positions, respectively.

cess and gradually decrease the reaching distance from the current position to the goal until the agent reaches the goal at $x_e$. The gradient descent method computes the direction at each move based on the function values within an infinitesimal area surrounding the current location, and thus the agent can only take an infinitesimally small step at each move.

However, in 2D maze, an agent usually takes a fixed, much larger step, e.g., one grid cell at each move. Therefore, we constrain that an agent moves by one grid cell along one of $2^3 = 8$ possible directions at each time, as shown in Fig. 2(a). We then query the learned implicit function for the reaching distance values at all possible positions that the agent can reach in one time step and move the agent to the position with the smallest reaching distance. We keep updating the agent's position until it reaches the goal (see Fig. 2(b)).

## 3.4 Conditional Environment Field

We now present how to generalize the environment field to arbitrary goal positions and scenes.
**Environment field for arbitrary goal positions.** To learn an implicit function that predicts different environment fields for different goal positions in the same scene, we utilize a conditional implicit function (Park et al., 2019) as shown in Fig. 3(b). The input to the implicit function is the concatenation of both the goal coordinates and the query position coordinates. The output is the reaching distance from the query position to the goal, i.e., environment field value. During training, we randomly sample a pair of empty grid cells and set one of them as the goal. We train the implicit function with the reaching distance computed by FMM.
**Environment field for arbitrary goal and environment.** To further extend the implicit function to an arbitrary environment, we propose a context-aligned implicit function as shown in Fig. 3(c). Given a scene context (i.e., a 2D maze map), we first extract scene context features by a fully convolutional environment encoder. Then, we forward the concatenation of (a) goal coordinates, (b) query position coordinates, and (c) scene context features aligned to the query position into the implicit

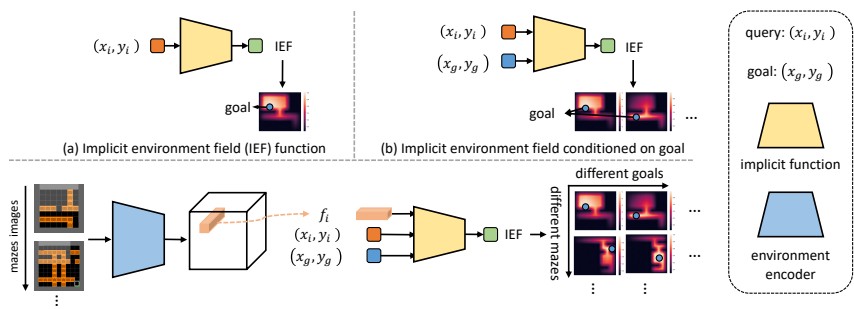

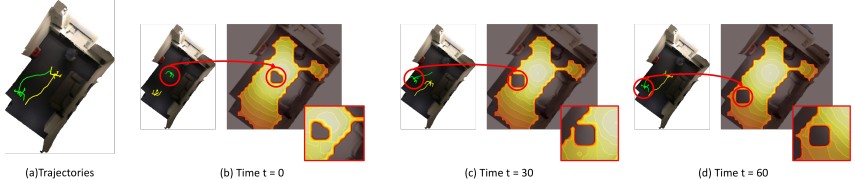

Figure 3: **Variants of implicit environment functions (IEFs).** The IEF in (a) assumes a fixed environment and a fixed goal position; (b) generalizes to different goal positions in the same environment; (c) further generalizes to different goal positions in different environments. Goal positions are represented by blue circles. In this figure, we use the 2D maze as an example, but models in (a) and (b) can be generalized to 3D scenes by simply changing the inputs from 2D coordinates to 3D coordinates.

Figure 4: **Bird's eye view navigation results.** We show the searched trajectories for two people in (a). In (b)-(d), we illustrate the positions of the two people at different time steps and how the position of one person (green) affects the reaching distance level-set of the other person (yellow) dynamically to avoid collision. Brighter color in (b), (c) and (d) indicates locations that are closer to the goal.

function and regress the reaching distance value for the query position. The fully convolutional environment encoder and scene environment feature alignment are two key ingredients in this design. The former enlarges the receptive field at the query position so that the implicit function has sufficient contextual information to predict the reaching distance to the goal. The latter encourages the implicit function to focus only on the query position while ignoring other scene context information. We note that hypernetworks (Sitzmann et al., 2020) are another option to encode arbitrary environments and goals. However, the hypernetwork solution is empirically less accurate than the proposed context-aligned implicit function, especially for mazes with larger sizes. Experimental comparisons between these two designs are presented in Section 5.2 and details of the hypernetwork are in Appendix C.1.

For the maze navigation task, we learn the context-aligned implicit function (Fig. 3(c)) using different mazes, along with the reaching distance computed by FMM as training data. During inference, we predict an environment field for an unseen maze with a given goal position and search for a feasible trajectory from any starting position to the goal using the strategy discussed in Section 3.3. Fig. 6 shows learned environment fields along with searched trajectories for unseen maze maps.

## 4 HUMAN NAVIGATION IN INDOOR ENVIRONMENTS

Going beyond the toy example of agent navigation in 2D mazes, we discuss how to apply the environment field to real-world 3D scenes in this section. Given the point cloud of each scene, we aim to generate a physically feasible and semantically plausible trajectory for a human from a starting position to the goal. We then align a random sequence of human poses onto the generated trajectory (see Fig. 7). While the *accessible* and *obstacle* locations in 2D mazes are well-defined, the *accessible region* for humans in 3D scenes is difficult to identify and usually requires prior knowledge of human behavior. We propose two solutions to determine the accessible region in 3D scenes: (a) A simple solution that utilizes the bird's eye view rendering of the 3D scene (see Section 4.1) and only models simple human motion, i.e., walking. (b) A learning-based solution that generates the accessible region in 3D scenes and models more complex human actions, e.g., sitting down, etc. (see Section 4.2).

### 4.1 BIRD'S EYE VIEW ENVIRONMENT FIELD

To model human walking, we define the floor area as the accessible region for humans from the bird's eye view of the 3D scene. We first produce a bird's eye view image and depth through rendering. Then we check the depth of each pixel in the bird's eye view image and label pixels with the maximum depth as the floor. All pixels belonging to the floor area are marked as *accessible* while other pixels are marked as *obstacles*. We then use this map as the scene environment condition. Similar to 2D maze navigation discussed in Section 3.4, we assume arbitrary goal positions in a dynamically changing indoor environment and learn a context-aligned implicit function shown in Fig. 3(c) to encode the environment field of the bird's eye view image. During inference, given the starting and goal positions on the floor, we search for a feasible trajectory that leads the human(s) to travel from the starting position to the goal while avoiding collision with obstacles in the room. We target both single and multiple human navigation.

**Single human navigation.** From a starting position, we assume an average human step size and move the human one step along one of the eight directions (see Fig. 2(a)) that yields the largest reduction in reaching distance. We repeat this process until the human reaches the goal. Given the searched trajectory in the bird's eye view, we align an existing human walking sequence onto it by rotating the human pose towards the tangent direction of the trajectory at each time step. More details of the alignment can be found in Appendix C.3. Fig. 1(f) and Fig. 4(a) show searched trajectories for humans from the bird's eye view together with the aligned human walking pose sequence.

**Multiple human navigation.** In addition to single human navigation, we show that our method can be easily applied to multiple people navigation – a challenging and computationally expensive task to be solved by any existing approach. Taking the two people in Fig. 4 as an example, we denote the yellow and green ones as person $P_1$ and $P_2$, respectively. To navigate $P_1$ from the left bottom corner to the center of the room while avoiding collisions with the furniture and $P_2$, at each time step, we modify the scene environment by marking the location of $P_2$ as an obstacle. Similarly, we update the scene environment for $P_2$ by marking the current position of $P_1$ as an obstacle and move $P_2$ accordingly. For this task, the scene environment is dynamically changing based on the motion of the other human and so is the predicted environment field. This is why we adopt the context-aligned implicit function in Fig. 3(c) rather than the model in Fig. 3(b). We also randomly mark empty spaces as obstacles during training to mimic the arbitrary scene context encountered during inference. Fig. 4(b)-(d) illustrate how the environment field of $P_1$ changes w.r.t. the motion of $P_2$, where the current position of $P_2$ is predicted as *inaccessible* to $P_1$ by the learned implicit function.

### 4.2 3D SCENE ENVIRONMENT FIELD

The solution discussed above simplifies human navigation in the 3D space to the 2D image space but is restricted to modeling the motion that mostly happens in the horizontal floor plane, i.e., walking. Next, we demonstrate how to generalize the environment field to 3D scenes and model more complex motions such as sitting and lying down, getting up, etc..

**Modeling data-driven accessible region in 3D scenes.** Unlike 2D mazes or the bird's eye view where the unoccupied floor region is explicitly given as the accessible region, in the 3D indoor scene, we define accessible region as plausible regions for human torsos to be placed. Given human pose sequences from the training set, we learn a VAE (Kingma & Welling, 2013) to model the distribution of human torso locations conditioned on the scene layout. As shown in Fig. 5, given a scene point cloud, we first take the global feature produced by the pooling layer of a PointNet (Qi et al., 2017) as the scene context feature. We map the context feature together with human torso location observations to a normal distribution using an encoder. A decoder then reconstructs the human torso location given the concatenation of a sampled noise from the normal distribution and the scene context feature. We learn this conditional VAE using the reconstruction objective on human torso locations together with the Kullback–Leibler (KL) divergence objective (Kingma & Welling, 2013). During inference, we sample random noise from a standard normal distri-

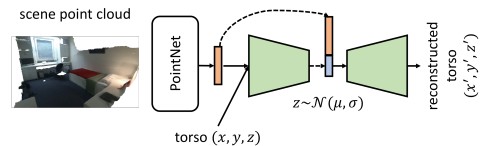

Figure 5: **Conditional VAE for accessible space generation.**

bution to generate feasible locations for human torsos to formulate the accessible region in a 3D scene. To further suppress the noisy generation by the VAE model and filter out locations that potentially lead to collisions with furniture in the room, we project all generated locations to the bird's

eye view and remove locations that fall on furniture or walls. Examples of generated accessible regions are shown in Fig. 7, where the VAE successfully predicts plausible locations suitable for humans to sit and walk on, etc. while avoiding collisions with furniture in the room.

**Environment field learning for 3D scenes.** Given the learned accessible region above, we define the environment field value of locations outside the accessible region as negative (i.e., inaccessible). We then model the field using the network in Fig. 3 (b), where the input to the implicit function is the concatenation of both the goal coordinates and the query position's coordinates. The output is the distance from the query position to the goal.

**Pose-dependent trajectory from a continuous field.** Given the starting and goal positions as well as a random sequence of human poses, we use the learned environment field to search a path and then align the human pose sequence onto the predicted path. Note that we focus on trajectory instead of human pose modeling, therefore we assume the action of the given pose sequence is consistent with the given starting and goal position. For instance, if given a random sequence of a human sitting down, the goal position should be on an object (e.g., soft, chair, etc.) that can afford the sitting action. Such a setting is commonly observed in applications such as motion matching (Holden et al., 2020). The key to navigating the human at each step is deciding its step length and direction. For direction computation, we assume the human can take one of the $3^3 = 27$ directions (denoted as $v_i, i = 1, 2, ..., 27$) in 3D space, similarly to the 2D maze scenario shown in Fig. 2, For step length, we compute the distance of adjacent torso locations in the given pose sequence as the step length denoted by $s$. Thus each possible location $l_i'$ human can reach within one step can be computed as $l_i' = l + v_i \times s$, where $l$ is the human's current location. Although the possible locations can be anywhere in the scene, the continuity of the learned environment field allows us to efficiently query the reaching distance for each possible location by feeding each $l_i'$ together with the goal's coordinates into the learned environment field. The human is then moved towards the location with the minimum reaching distance. This process is repeated until the human reaches the goal position. We emphasize that different from the existing human motion prediction method (Cao et al., 2020), the above trajectory prediction process can be carried out given arbitrarily far away starting and goal positions since the environment field densely covers the entire scene.

**Path optimization by affordance.** The path searching process discussed above can be further optimized by taking affordance into consideration at each step. For each possible location the human can reach within one step size, we check if the human is well supported and does not collide with other objects at these locations. We move the human to the location with the smallest reaching distance value while obeying these two constraints. Similar to (Li et al., 2019), to determine if a pose is well supported, we check if the torso, the left lap, and the right lap joints of sitting poses, as well as the feet joints of standing poses have non-positive signed distances to the scene surface. To determine if a pose collides with other objects, we check if all joints of the pose except the aforementioned support joints have non-negative signed distances. Finally, we align the pose sequences onto the searched path, details of this alignment process are discussed in Appendix C.3. Fig. 7(b), (d) and (e) illustrate the searched trajectory along with the aligned human poses.

## 5 EXPERIMENTS

### 5.1 EXPERIMENTAL SETTINGS

We train the conditional VAE described in Section 4.2 and all implicit functions using the Adam solver (Kingma & Ba, 2014) with a learning rate of $5 \times 10^{-5}$. To balance the KL-divergence and the reconstruction objectives in the VAE, we adopt the Cyclical Annealing Schedule introduced in (Fu et al., 2019). We evaluate the proposed environment field on agent navigation in 2D mazes (Section 5.2) and compare with the VIN (Tamar et al., 2016). For dynamic human motion modeling in 3D scenes (Section 5.3), we compare with the HMP (Cao et al., 2020) and continuous sampling-based methods RRT (Bry & Roy, 2011) and PRM (Kavraki et al., 1996). We note that VIN is based on Convolutional Networks and cannot be easily extended to 3D scenes.

### 5.2 AGENT NAVIGATION IN 2D MAZES

We evaluate the proposed environment field on the Minigrid (Chevalier-Boisvert et al., 2018) and Gridworld maze datasets (Tamar et al., 2016), which include 2D mazes with randomly placed obstacles (see Fig. 6(a)). Fig. 6(b) and (c) show the learned environment fields and the level sets (each level set consists of locations of the same reaching distance to the goal) respectively. Each environment field accurately encodes the reaching distance from any *accessible* point to the goal while

Table 1: **Ablation study on network architecture**. The $11 \times 11$ mazes are from the Minigrid (Chevalier-Boisvert et al., 2018) dataset while all other mazes are from the Gridworld (Tamar et al., 2016) dataset. *IF* stands for implicit function.

| Maze Size | 8×8 | 11×11 | 16×16 | 28×28 |
|---|---|---|---|---|
| Hypernetworks | 98.08 | 82.05 | 20.77 | 3.13 |
| Context-aligned IF | 99.69 | 80.14 | 98.29 | 97.00 |

Table 2: **Success rate** and **efficiency** for maze navigation.

| Size | Success rate↑ | | | Efficiency (*milliseconds*)↓ | | |
|---|---|---|---|---|---|---|
| | 8×8 | 16×16 | 28×28 | 8×8 | 16×16 | 28×28 |
| VIN | 99.60 | 99.30 | 96.97 | 3.88 | 22.72 | 82.07 |
| Ours | 99.69 | 98.29 | 97.00 | 13.26 | 14.81 | 20.41 |

successfully detecting all *obstacles* in the maze. Figure 6(a) demonstrates the effectiveness of using the learned environment field in navigating an agent to the goal while avoiding collision with obstacles. Quantitatively, we use the success rate – the percentage of paths that successfully lead an agent to the goal among all searched paths, as the evaluation metric.

**Hypernetworks *v.s.* context-aligned implicit function.** Table 1 shows the ablation study on the network architectures discussed in Section 3.4. The context-aligned implicit function performs comparably to hypernetworks on smaller-sized mazes ($8 \times 8$ and $11 \times 11$ mazes). However, hypernetworks fail on larger-sized mazes ($16 \times 16$ and $28 \times 28$ mazes), demonstrating the superiority of context-aligned implicit functions over hypernetworks.

**Comparison with VIN.** We compare the proposed method with VIN (Tamar et al., 2016) on the Minigrid dataset (Chevalier-Boisvert et al., 2018). We use the same train and test split as VIN, and learn different implicit functions for mazes of different sizes. Table 2 shows the suc-

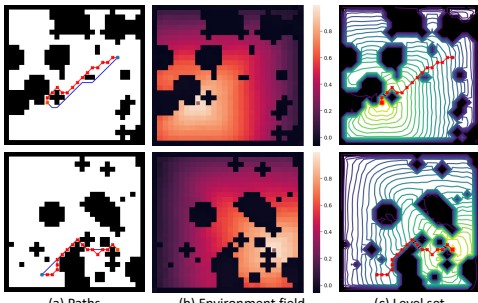

(a) Paths    (b) Environment field    (c) Level set

Figure 6: **Grid-world maze navigation.** Blue paths in (a) are planned by FMM while red paths are searched by the learned environment field. Green circles and orange squares represent starting positions and goal positions. In (b)(c), a brighter area or contour is closer to the goal.

cess rate and average time taken to search a path by different methods. The proposed method achieves a comparable success rate with much less time in larger-sized mazes compared to VIN. This is because the VIN requires a network forward pass at each location to decide the next optimal step, while the proposed method only needs one network forward pass to obtain the reaching distance values of all locations in the maze and can guide the agent at any location to reach the goal.

## 5.3 INDOOR HUMAN NAVIGATION MODELING

**Bird's eye view navigation.** In Fig. 4(b)-(d), we illustrate how the current position of one person affects the environment field of the other person. First, the learned level set effectively encodes the reaching distance between any point to the goal position. Areas that are closer to the goal have lower reaching distance values and vice versa. Second, the learned environment field dynamically changes based on the position of the other person. For instance, in Fig. 4(b), the current location of the person colored in green is successfully predicted as *inaccessible* by the other person, i.e., it has a large reaching distance value. These results show that the learned implicit function can effectively model the environment field in a scene and generalize to a dynamically changing environment.

**Data-driven human accessible region learning.** We show qualitative results of the generated accessible region (see Section 4.2) in Fig. 7(b). The generated accessible region uniformly spreads out in the 3D room and is plausible for all kinds of human actions. For instance, the points on the chair and sofa are feasible locations for sitting human torsos, while the points in midair are plausible locations for walking human torsos. Besides generated accessible region, we show the learned environment field in Fig. 7(a), where the closer a point is to the goal, the smaller its reaching distance value is (brighter point).

**Human trajectory prediction.** We search a trajectory for a random human pose sequence (see Section 4.2) in Fig. 7(c) and (d). The trajectory successfully avoids colliding with furniture in the room while leading the human towards the goal. We emphasize that the starting and goal positions are 120 frames (8 seconds) apart, showing the capacity of the proposed method in predicting long-term dynamic human motion (see the video in the supplementary material). Our model outperforms

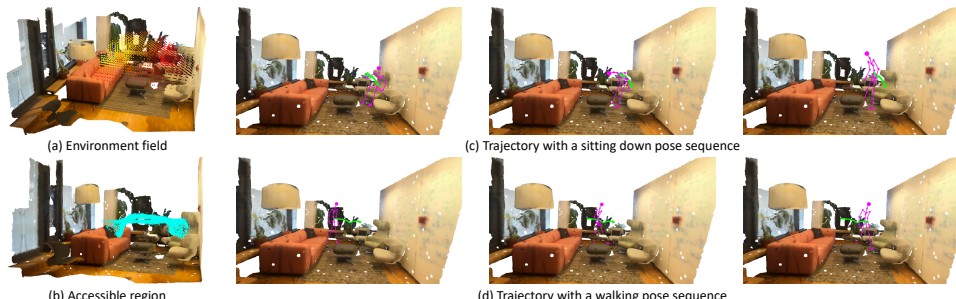

Figure 7: **Dynamic human motion modeling on PROX (Hassan et al., 2019)** The accessible region in (b) includes generated torso locations by the VAE model discussed in Section 4.2. In the environment field in (a), the brighter a point is, the closer it is to the goal.

Table 3: Quantitative comparison with RRT (LaValle, 1998) and PRM (Kavraki et al., 1996) on the PROX dataset. "AFF" indicates the pose dependent search discussed in Section 4.2.

| Method | RRT | PRM | Ours (w/o AFF) | Ours |
|---|---|---|---|---|
| distance ↓ | 0.127 | 0.139 | **0.015** | 0.015 |
| time (s) ↓ | 0.153 | 1.310 | **0.028** | 0.038 |

the state-of-art method (Cao et al., 2020) that can only effectively capture human motion within 2 seconds. Furthermore, the human poses are plausible at each location on the trajectory, demonstrating the effectiveness of the pose-dependent trajectory search process discussed in Section 4.2.

**Comparison with RRT (LaValle, 1998) and PRM (Kavraki et al., 1996).** We present quantitative comparisons with sampling-based path planning methods in Table 3. We evaluate two metrics: (a) distance between the end points of a searched path and the goal. (b) average time cost for a single trajectory search. The number of sampling points and nearest neighbors in PRM are set to 500 and 5. For RRT, the maximum iteration for trajectory search is 500. More details can be found in Appendix C.6. Overall, our method is more efficient thanks to the continuous environment field, from which the next step can be predicted by a single fast network forward pass. In addition, our method yields better trajectories that successfully reach the goal as shown in Table 3.

**Comparison with HMP (Cao et al., 2020).** Since we focus on trajectory prediction, we quantitatively evaluate with the PathNet in the HMP using the distance metric discussed above as well as the ratio of valid poses to all poses on the trajectory. As described in Section 4.2, we define a pose as valid if it is both well supported and collision-free in the scene. We use the same train and test trajectories as the HMP method, with each trajectory including 30 frames. We note that the PathNet in the HMP requires torso locations in the first 10 frames of each trajectory as input and predicts the torso locations for the following 20 frames. While our method directly predicts 30 locations that lead the human to the goal for the entire 30 frames. Quantitative results in Table 4 show that our method is effective for navigating humans towards goal positions. Furthermore, after taking affordance into consideration (see Section 4.2), our model is able to better fit the human poses to the scene, as shown in the last column of Table 4.

Table 4: Quantitative evaluations of human motion trajectory prediction on the PROX dataset. "AFF" is the same as in Table 3. "Support" and "free" indicate the ratio of poses that are either well supported or collision-free, respectively. "Valid" indicates the ratio of poses that obey both constraints.

| Method | distance ↓ | support ↑ | free ↑ | valid ↑ |
|---|---|---|---|---|
| HMP (Cao et al., 2020) | 0.019 | 81.40 | 72.34 | 65.00 |
| Ours (w/o AFF) | 0.015 | 84.47 | 75.43 | 71.36 |
| Ours | **0.015** | **86.22** | **78.56** | **75.32** |

# 6 CONCLUSION

In this paper, we propose the environment field that encodes the reaching distance between a pair of points in either 2D or 3D space, with implicit functions. The learned environment field is a continuous energy surface that can navigate agents in 2D mazes in dynamically changing environments. We further extend the environment field to 3D scenes to model dynamic human motion in indoor environments. Extensive experiments demonstrate the effectiveness of the proposed method in solving 2D mazes and modeling human motion in 3D scenes.

ACKNOWLEDGMENTS

Ming-Hsuan Yang is supported in part by the NSF CAREER grant 1149783.

REPRODUCIBILITY STATEMENT

We describe the proposed framework in Fig. 3, with training and implementation details described in Section C.5 in the Appendix. Our method is evaluated on publicly avilable dataset (Chevalier-Boisvert et al., 2018; Tamar et al., 2016; Hassan et al., 2019). We plan to release the code.

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

## A  OVERVIEW

In this appendix, we provide more details and experimental results of the proposed implicit environment field approach. First, we differentiate the proposed method and VIN (Tamar et al., 2016) in Section B. Then, we discuss the implementation details of our method and baselines in Section C. Finally, more experimental results, including failure cases are shown in Section D.

## B  COMPARISON WITH VIN

Our method is fundamentally different from VIN (Tamar et al., 2016) in the following aspects: a) Goal. We aim to utilize scene understanding to model dynamic agent behaviors, especially human behaviors in indoor environments. However, VIN aims at learning neural networks to approximate the value-iteration algorithm and focuses on learning reward, transition and value functions to obtain the optimal policy for agent navigation. b) Capacity. Our method is more powerful than VIN and other reinforcement learning methods. For instance, it naturally enables navigating multiple humans in the same scene, which is hard to achieve using VIN if corresponding supervision is not provided. c) Flexibility. We utilize implicit functions – a representation of a 3D scene to model the inherent property (reaching distance) of the scene, which captures how long it takes to travel from a point to a given goal plausibly and feasibly. Thanks to such a representation, our method can be flexibly generalized to complex continuous scenes like 3D indoor environments. While VIN depends on convolution operations, which cannot be directly applied to continuous 3D scenes.

Comparisons with other path planning methods can be found in Section 2.

## C  IMPLEMENTATION DETAILS

### C.1  HYPERNETWORKS

To encode different maze environments as well as goal positions, we use a hypernetwork (Ha et al., 2016) as shown in Fig. 8. For each maze map, we first use a hypernetwork (blue trapezoid in Fig. 8) to predict the parameters of a hyponetwork (yellow trapezoid in Fig. 8). The hyponetwork takes the concatenation of the goal coordinates as well as the query position coordinates and outputs the reaching distance between them. The hypernetwork includes 11 convolutional layers followed by 7 fully-connected layers. The hyponetwork contains 7 fully-connected layers, each of which is followed by a periodic activation as in (Sitzmann et al., 2020). A comparisons between the hypernetwork architecture and the proposed context-aligned implicit function can be found in Section 5.2.

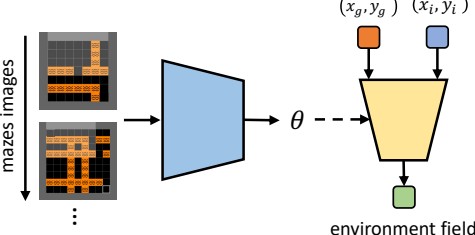

Figure 8: **Hypernetwork architecture.** Blue trapezoid is the hypernetwork that predicts the parameters of the hyponetwork (the yellow trapezoid).

### C.2  TRAJECTORY SEARCHING IN 2D MAZES

The agent in a 2D maze may be stuck at a location during the trajectory searching process. To resolve this issue, we sample from all possible locations in the next step, instead of directly taking the one with the smallest reaching distance. Specifically, for all eight possible directions, we compute a probability vector formed as the inverse of their reaching distance. We treat this probability vector as a multinomial probability distribution and sample one direction out of eight.

## C.3 TRAJECTORY SEARCHING IN THE BIRD'S EYE VIEW

**Multiple humans navigation.** Here we summarize the algorithm to navigate multiple humans simultaneously while avoiding collision with objects or other humans, as discussed in Section 4.1. As shown in Algorithm 1, for each human $i$ at each time step $t$, we mark the current positions of other humans $j$ as obstacles and search the optimal step based on the modified scene environment. We denote the total step number as $T$ and total human number as $H$.

---

**Algorithm 1** Multiple humans navigation.

---

$t \leftarrow 1$
$i \leftarrow 1$
$j \leftarrow 1$
**while** $t \leq T$ **do**
    **while** $i \leq H$ **do**
        Initialize scene environment
        **while** $j \leq H$ **do**
            **if** $i \neq j$ **then**
                Mark the position of human $j$ as an obstacle
            **end if**
        **end while**
        Predict environment field for human $i$ w.r.t the modified scene environment
        Search next optimal step for human $i$
    **end while**
**end while**

---

**Aligning walking sequence to searched trajectories.** As discussed in Section 4.1, we align a random human walking sequence from the SURREAL dataset (Varol et al., 2017) to the searched trajectory from the bird's eye view. We show this process in Fig. 9. At each step, we compute the angle $\theta$ between the human torso displacement (red arrow) and the trajectory's tangent (green arrow). The human pose is then rotated by $\theta$ to align with the trajectory's direction.

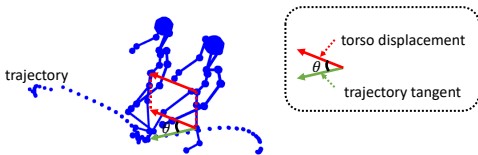

Figure 9: **Aligned walking sequence to searched trajectory.** The figure is for visualization purposes only. The human takes a much smaller step in reality.

## C.4 TRAJECTORY SEARCHING IN 3D INDOOR ENVIRONMENT

**Aligning pose sequence to searched trajectories.** Given the predicted trajectory in a 3D room (see Section 4.2), we align a random walking or getting-up pose sequence from the SAMP dataset (Hassan et al., 2021) onto the trajectory. The alignment process of a walking sequence is the same as in the bird's eye view scenario discussed in Section C.3. While for the getting-up pose sequence, we first translate each pose by moving its torso onto the predicted trajectory. To enforce the poses to be well afforded by its surroundings, we adjust it by rotating within [-6, 6] degrees and choose the angle with the best affordance (i.e. with the most joints well supported and collision-free). Note that the fitting process is the same for trajectories predicted by our method and the baseline method (Cao et al., 2020) for fair comparisons.

## C.5 NETWORK TRAINING

In this work, instead of using the raw reaching distance computed by FMM as supervision to learn the implicit function, we make two changes: (a) We use the reciprocal of the reaching distance from the feasible point to the goal and normalize it to $[0, 1]$. (b) We encourage the implicit function to predict negative one instead of a extremely small values for locations occupied by obstacles. In this way, the network can better differentiate feasible locations and obstacles.

During the training of the VAE model discussed in Section 4.2, besides the points on the training human trajectories, we also augment more points in the room. Specifically, we use a random standing human pose and include all locations that can afford this pose, i.e., the pose can be supported by the floor and does not collide with other objects.

We learn a single conditional VAE model for all training trajectories in different scenes as in (Cao et al., 2020), while a separate implicit function is trained for each 3D indoor scene. All models introduced in this paper are trained with the Adam solver (Kingma & Ba, 2014) using a learning rate of $5 \times 10^{-5}$. We implement the proposed method in the PyTorch framework (Paszke et al., 2019).

## C.6 PATH PLANNING BY RRT AND PRM

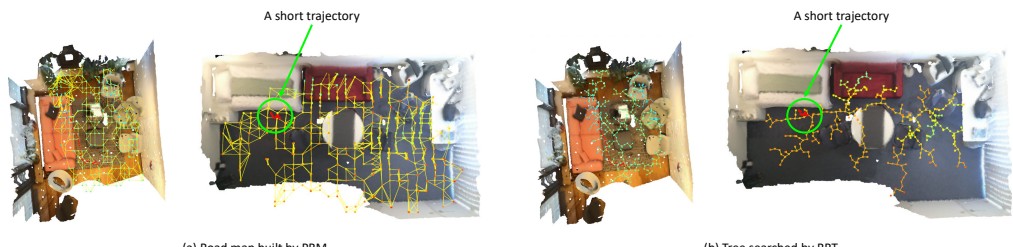

Figure 10: **Visualization of path planning by RRT (Bry & Roy, 2011) and PRM (Kavraki et al., 1996).** Ground-truth trajectories are colored in red.

For path planning, the RRT (Bry & Roy, 2011) method starts from a given point and builds a tree by adding randomly sampled collison-free points to the tree until one branch of the tree reaches close to the goal. Instead, the PRM (Kavraki et al., 1996) first samples a set of collision-free points in the space and builds a map by establishing edges wherever possible. It then searches a trajectory from a given point to the goal using the Dijkstra's algorithm (Dijkstra, 1959). Both these methods are less efficient than the proposed method due to the randomness in the sampling process. As shown in Fig. 10, the two methods build a large road map or tree even when searching for a short trajectory (green circles). On the contrary, the proposed method efficiently computes the next optimal step with one network forward pass and thus is more efficient. A quantitative comparison can be found in Table 3.

# D    EXPERIMENTAL RESULTS

## D.1    MORE 2D MAZE NAVIGATION RESULTS

In Fig. 11, we show more results of agent navigation in 2D mazes as well as learned environment fields. As discussed in Section 5.2, the learned environment field successfully captures the reaching distance from any point to the goal (see Fig. 11(b)&(c)) and effectively guides the agent from the starting position to the goal in Fig. 11(a).

## D.2    DATA-DRIVEN HUMAN MOTION MODELING

In Fig. 13, we show more results of fitting given human pose sequences to searched trajectories in 3D indoor environments.

As discussed in Section 4.2, given a human pose sequence, we compute the step size of the human at each time step and dynamically determine the best next move w.r.t the step size. We also utilize the given human pose to avoid moving humans to locations that violate physical constraints in the room, i.e., locations that cannot well support the human or lead to collision with other objects.

## D.3    FAILURE CASES

Since we only focus on predicting human trajectories using the learned environment field in this work, we fit a given random pose sequence onto the predicted trajectory rather than model human

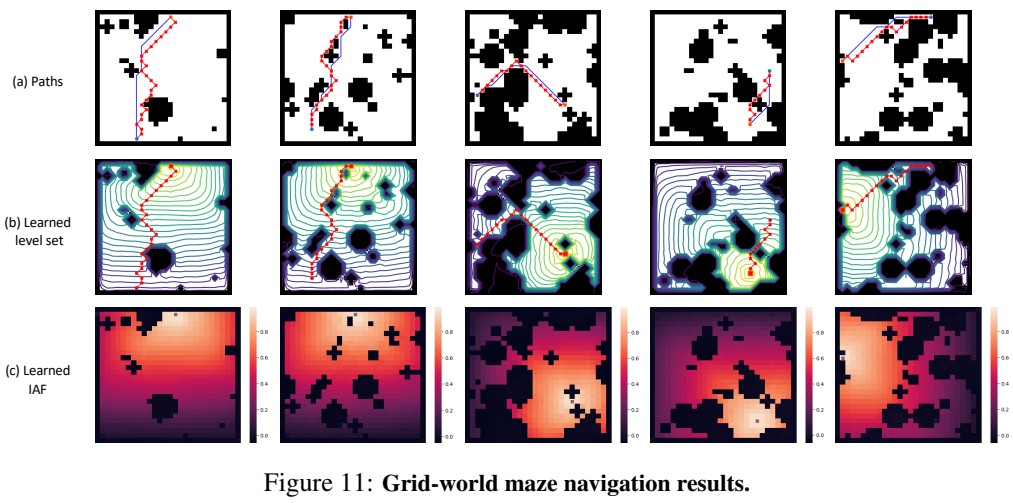

Figure 11: **Grid-world maze navigation results.**

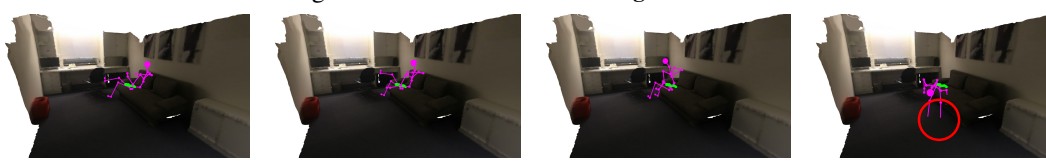

Figure 12: **Failure cases.**

pose distribution. Although we assume the action of the given human pose sequence is consistent with the start and goal position (e.g. fit a getting-up sequence to a trajectory ending at a sofa's surface), however, the given pose sequence may still not well suite for the scene. For instance, The red circle shown in Fig. 12 demonstrates a failure case where the human collides with the floor. This collision is mainly cased by the fact that the given "getting-up" sequence does not suite the height of the sofa. We believe a possible solution is to further utilize the environment field to predict human poses besides trajectories. We leave further exploration along this line to future works.

## D.4 TRAJECTORY IN 3D SCENES VISUALIZATION

In Fig. 14, we visualize the trajectories searched by our method and the baseline method Cao et al. (2020). As shown in the figure, our method produces trajectories that include less noise and align better with the ground truth trajectories better.

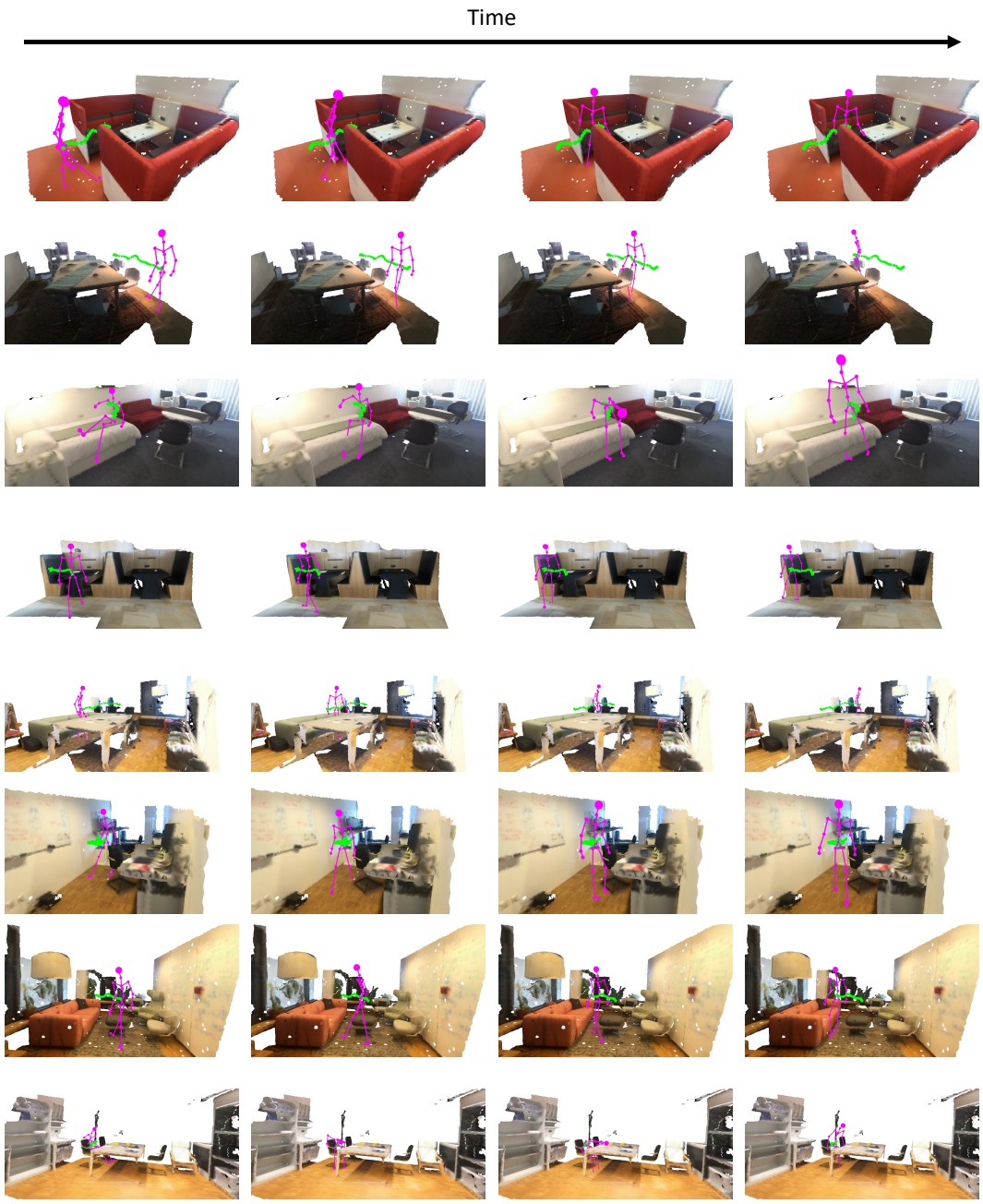

Figure 13: **Dynamic human motion modeling on PROX (Hassan et al., 2019).** We highly recommend viewing the accompanying video for dynamic visualizations.

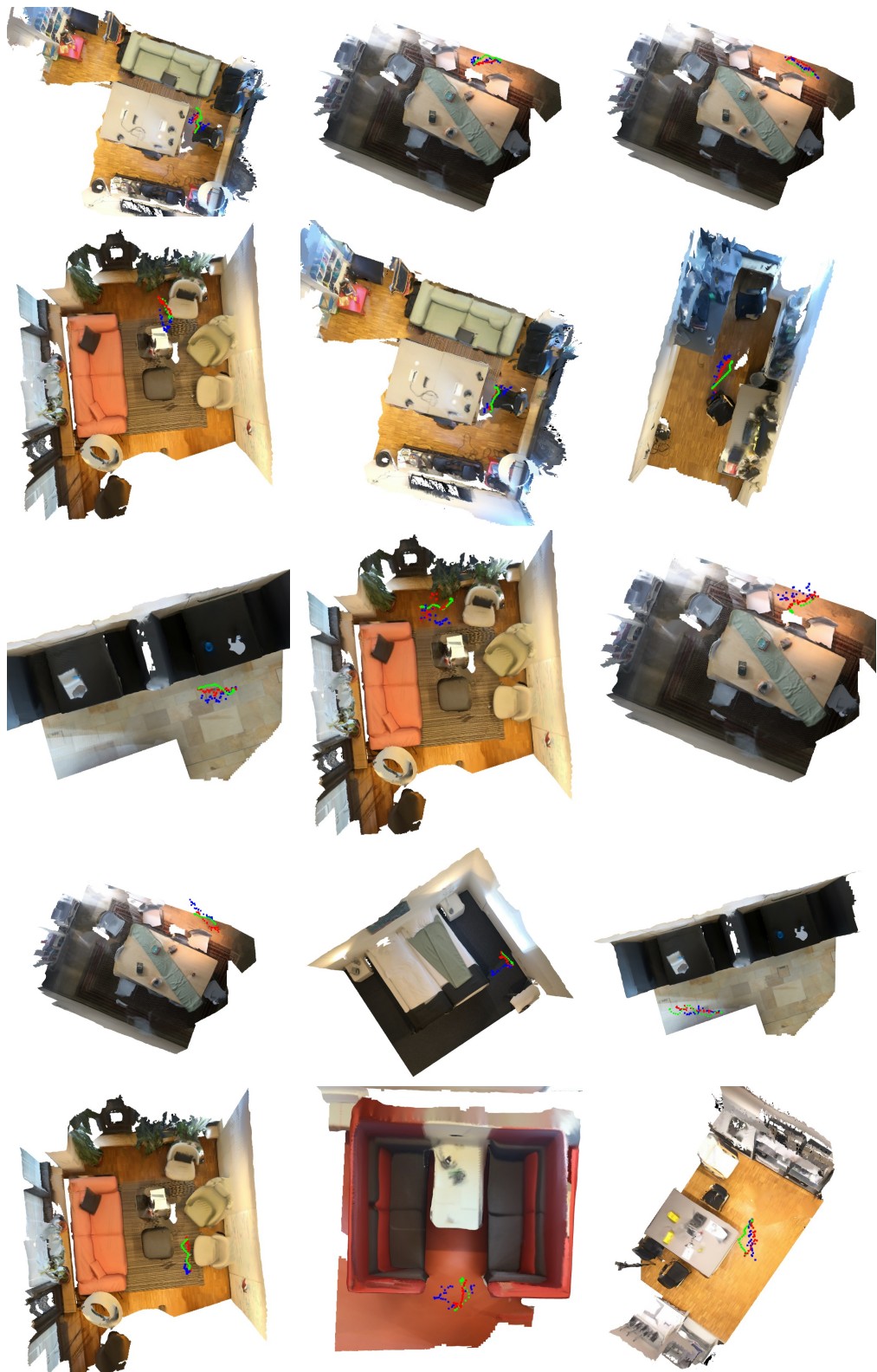

Figure 14: **Trajectory visualization in 3D scenes.** The ground truth trajectory, the trajectory searched by HMP Cao et al. (2020) and the trajectory searched by our algorithm are shown in red, blue and green.

