# OpenReview forum: "Learning Continuous Environment Fields via Implicit Functions"
_ICLR.cc/2022/Conference — ICLR 2022 Poster_

### Official Review · Reviewer_z3Y4 · 2021-10-31

**Correctness:** 1
**Technical Novelty And Significance:** 1
**Empirical Novelty And Significance:** 1
**Recommendation:** 1
**Confidence:** 5

**Main Review:**

While the general idea, having a neural network learn to reproduce the output of an A* or Dijkstra algorithm is sensible, what is proposed in this paper is simply not comprehensible.

The issues start with it being unclear whether or not the idea is to reproduce something akin to a Dijkstra algorithm, i.e., any point to any point, or an A* algorithm, i.e., any point to a single target location. This confusion is caused by the ambiguous writing used in Figure 1 and the abstract, implying Dijkstra and the introduction, which implies A*. The related work also lacks various aspects, such as voxelgrid and tsdf representations in the scene representation section or potential field methods and rmps in the path planning aspect. The affordance prediction section also seems to be a misnomer as it focuses on trajectory prediction, which is different from affordances.

The description of the method itself is simply devoid of all required detail, such as architecture, training regime, cost function regularization, biases induced due to the discretized nature of the training data, etc. If one were to attempt to reproduce the method based on the information provided, it would be impossible to know where to start and whether one succeeded in reproducing what is described.

The section that describes the human motion prediction aspect is devoid of the required detail, just as the previous section. Furthermore, the general idea being described would likely fall into a sort of motion planning approach rather than a motion prediction method.

In many instances, the specifics of the technical detail provided appears wrong or not the commonly used form. For example, the definition of the sdf values is the opposite of what is typically used in TSDF representations. Another issue is that the paper seems to imply that RRT-like methods work in discretized representations only, which is not the case.

The experiments highlight further problems with the proposed method. While in earlier parts, the paper describes the importance of a continuous representation, all experiments are performed on discretized grid environments. Why is this? Additionally, neither the methods used in the comparison nor the metrics used are described, and the experimental setup details remain vague. Where there is information, it raises more questions. In the RRT comparison section, one of the metrics is described as "distance between the endpoints of a searched path and the goal," yet the RRT method is claimed to achieve a value > 0 in these tasks. As an RRT is guaranteed to find a solution if one exists and creates a fully connected path between the start and goal location, this value can not be anything other than 0. As such, it is not clear what this metric truly measures nor that it is informative.

There is also nowhere an evaluation of the optimality or safety of the resulting paths. This is something for which theoretical information exists for RRTs, but with neural networks, especially those having the interpolate and perform out-of-domain predictions is anything but guaranteed and a big concern.

**Summary Of The Paper:**

The paper proposes a method to learn an environment field that predicts the distance from any location in the map to a query location. The method builds on implicit surface representation ideas and extends it with a few other aspects for motion prediction.

**Summary Of The Review:**

While the general idea is viable and something being researched, what is being proposed in this paper is simply too unclear and vague to be assessed.

---

> ### Author Response · Authors · 2021-11-22
> **For Reviewer z3Y4 (Part II)**
>
> > **Q:** *The definition of the sdf values is the opposite of what is typically used in TSDF representations*
>
> **A:** We define the SDF value following the definition used in neural implicit functions [1], our work has nothing to do with scene TSDF. As we mentioned before, a neural implicit function is a neural network which takes a location as input and outputs the signed distance for that location. Please refer to [1] and our related work to check recent literature on this topic if the reviewer is not familiar with it.
>
> [1] Sitzmann et al. Implicit neural representations with periodic activation functions. NeurIPS'2020.
>
> > **Q:** *Another issue is that the paper seems to imply that RRT-like methods work in discretized representations only, which is not the case.*
>
> **A:** We respectfully disagree with the reviewer that we have implied RRT-like methods work in discrete representations only. On the contrary, we used RRT and RPM as our baselines in continuous scenes like 3D rooms. We demonstrate qualitative results of RRT for 3D rooms in Fig. 10 in the Appendix and quantitative results in Table 3 in the originally submitted manuscript.
>
> > **Q:** *While in earlier parts, the paper describes the importance of a continuous representation, all experiments are performed on discretized grid environments. Why is this?*
>
> **A:** We respectfully disagree. Grid environment, i.e., 2D maze, is just one of the many experiments. Our experiments on predicting human motion in 3D scenes are all based on our continuous representation.
> Based on the reviewer’s questions on human motion prediction, the reviewer seems to be aware that there are experiments beyond grid environments. But here the reviewer states all experiments are performed on grid environments. **The reviewer seems to self-contradict** to the authors’ point of view.
> Please see the General Response for more discussions on experiments with continuous representation.
>
> > **Q:** *Additionally, neither the methods used in the comparison nor the metrics used are described, and the experimental setup details remain vague.*
>
> **A:** We respectfully disagree with the reviewer that we did not discuss metrics and baseline methods. We describe them very explicitly in the paper:
> - We describe the baseline for 2D maze (i.e., VIN) in the third paragraph in Section 5.2. The metric (successful rate) is described by the last sentence in the last sentence of paragraph 1 in Section 5.2.
> - We describe the baseline for 3D scenes (i.e., RRT, PRM, HMP) as well as the evaluation metric (distance to the goal position, time, valid pose ratio) in the last two paragraphs in Section 5.3.
>
> We again urge the reviewer to look carefully into our paper in the sections mentioned above.
>
> > **Q:** *There is also nowhere an evaluation of the optimality or safety of the resulting paths. This is something for which theoretical information exists for RRTs, but with neural networks, especially those having the interpolate and perform out-of-domain predictions is anything but guaranteed and a big concern.*
>
> **A:** As stated in our abstract, introduction, and other parts throughout the paper, our contribution is introducing a novel scene representation that can be applied to agent navigation and human behavior modeling. It is **NOT to reproduce any path planning method**. Thus the theoretical proof on optimality and safety accompanied with RRTs do not apply here.
> Nevertheless, empirically, we did evaluate the generated trajectories by counting human collision and support score as shown in Table 4, this demonstrates how accurately the paths model human behaviors.
> While theoretical work is important, our main focus is on empirical study of a new representation, which we argue should not be neglected.
>
> > **Q:** *In the RRT comparison section, one of the metrics is described as "distance between the endpoints of a searched path and the goal," yet the RRT method is claimed to achieve a value > 0 in these tasks. As an RRT is guaranteed to find a solution if one exists and creates a fully connected path between the start and goal location, this value can not be anything other than 0.*
>
> **A:** The RRT algorithm guarantees to reach the goal given infinitely many sampling steps, which is not possible in real world application. As discussed in the fifth paragraph (“Comparison with RRT”) of Section 5.3 (originally submitted paper), we set the maximum iterations of the RRT to 500. This already takes five times (0.153s) longer than the proposed method (0.028s) as shown in Table 3.

---

> > ### Comment · Reviewer_z3Y4 · 2021-11-26
> > **Reply**
> >
> > The following is the feedback on the detailed rebuttal by the authors for the main points only which influenced the score rather than minor points of
> >
> > Reaching distance points: My point was not about what the actual definition of that reaching distance aims to be. Is it between any two points in the environment (N to N) or is it between a specific point any other position in the scene (1 to N)?
> >
> > Lack of technical detail: The comment is about technical detailed information not the absence of any information. Yes, the paper provides general ideas of what components are used but neither their choice is discussed, why things are used, nor the specifics, e.g. layer depths, layer count, activation functions, etc. Stating that an architecture follows another one simply doesn't provide enough detail so another researcher could be confident they would implement the method as described.
> >
> > The detail in comparisons: I assume that the third paragraph in Section 5.2 refers to the one starting with "Comparison with VIN"? If so, I do not see how that text describes the method other than referring to its paper. While the metric used in Table 1 is described in general the description remains textual and is thus open to misinterpretation. There are also some odd placements of these descriptions, for example, the description of some metrics used in Table 4 appears only in the last paragraph of 4.2. While this might be due to space constraints it is not friendly to a reader.
> >
> > Safety & RRT: I do not dispute that the paper's main goal is the introduction of an environment representation, however, if comparisons are made on specific tasks to showcase its utility the concerns that arise in such tasks need to be taken into account. I have no issue if the proposed method not being able to provide a theoretical guarantee or would end up having more collisions than a method designed to be good at those aspects. But in path planning tasks that is a critical aspect, as short trajectories that have a high chance of generating collisions are not solving the problem statement, so to speak. A similar argument goes for the RRT being used with just 500 steps. Yes the NN can be much faster, and so having a limit on the number of steps is sensible, however, it would have been good to showcase the amount of time required by an RRT to find the optimum. This would not reduce the value of the proposed method but simply demonstrate the difference and tradeoff available. On a side note, based on the citation the initial version of RRT and PRMs were used instead of more recent variants? I don't expect the results to change drastically if a bi-rrt* or similar had been used but it might be valuable to do so, to avoid being seen as an unfair baseline selection.
> >
> > Overall, while I appreciate the comments of the authors no novel information has been provided in the critical aspects, other than stating that they disagree with my assessment of the proposed text. While this is fine and understandable, after checking the aspects referenced by the authors and reading over certain paragraphs again, my overall assessment did not change.

---

> > > ### Author Response · Authors · 2021-11-30
> > > **Further Clarification (Part I)**
> > >
> > > > **Q:** *Reaching distance points: My point was not about what the actual definition of that reaching distance aims to be. Is it between any two points in the environment (N to N) or is it between a specific point any other position in the scene (1 to N)?*
> > >
> > > **A:** We describe and emphasize our goal: output a field for any two points, any environments, even in the first sentence of the abstract. Our Section 3 is all about how to achieve this goal. In detail, we discussed:
> > >
> > > - Any point to a single target in a single environment setting in Section 3.2.
> > > - Any point to any point in a single environment in the second paragraph of Section 3.4.
> > > - Any point to any point in different environments in the third paragraph in Section 3.4.
> > > All the above settings are clearly visualized in Fig.3, (a)-(c).
> > >
> > > > **Q:** *Lack of technical detail: The comment is about technical detailed information not the absence of any information. Yes, the paper provides general ideas of what components are used but neither their choice is discussed, why things are used, nor the specifics, e.g. layer depths, layer count, activation functions, etc. Stating that an architecture follows another one simply doesn't provide enough detail so another researcher could be confident they would implement the method as described.*
> > >
> > > **A:** At this point, we are confused about what the reviewer's comments really request. In the initial reviews, reviewer z3Y4 comments as: *"The description of the method itself is simply devoid of all required detail, such as architecture, training regime, cost function regularization, biases induced due to the discretized nature of the training data, etc."*. After we pointed out to all these details that are already included in the **submission**, now reviewer z3Y4 is asking for *"layer depths, layer count, activation functions, etc,"*. We would like to emphasize that this paper is about learning the environment field rather than designing and training neural networks. There's no point in discussing details such as whether we use ReLU or LeakyReLU and how that would influence the framework, especially when we promise to release the code. Also, *"stating that an architecture follows another one"* points a reader to the reference that provides detailed information about the architecture, this is exactly the functionality of reference, otherwise we would have to start the paper by introducing the concept of convolution.
> > >
> > > > **Q:** *The detail in comparisons: I assume that the third paragraph in Section 5.2 refers to the one starting with "Comparison with VIN"? If so, I do not see how that text describes the method other than referring to its paper. *
> > >
> > > **A:** Again, it is not our responsibility to describe a baseline method in detail in our paper, one can refer to the reference for more details of the baseline method. Instead, we have differentiated the proposed method from the VIN in details in the related work, in Appendix B. If the reviewer is interested in the baselines, please refer to the cited papers.
> > >
> > > > **Q:** *While the metric used in Table 1 is described in general the description remains textual and is thus open to misinterpretation.*
> > >
> > > **A:** We would like the reviewer to be more specific about the *"misinterpretation"*, otherwise we do not understand why *"description remains textual and is thus open to misinterpretation"* and what the reviewer is actually *"misinterpreting"*.
> > >
> > > > **Q:** *There are also some odd placements of these descriptions, for example, the description of some metrics used in Table 4 appears only in the last paragraph of 4.2. While this might be due to space constraints it is not friendly to a reader.*
> > >
> > > **A:** Please be specific where to put the description so that it is friendly to a reader.

---

> > > ### Author Response · Authors · 2021-11-30
> > > **Further Clarification (Part II)**
> > >
> > > > **Q:** Safety & RRT: I do not dispute that the paper's main goal is the introduction of an environment representation, however, if comparisons are made on specific tasks to showcase its utility the concerns that arise in such tasks need to be taken into account. I have no issue if the proposed method not being able to provide a theoretical guarantee or would end up having more collisions than a method designed to be good at those aspects. But in path planning tasks that is a critical aspect, as short trajectories that have a high chance of generating collisions are not solving the problem statement, so to speak.
> > >
> > > **A:** We believe the reviewer misunderstands the basic idea of this paper. **Our approach does model the collision**: unlike traditional path searching algorithm, e.g., RRT, that detects obstacles during searching, our model **represents them within the learned field**, i.e., areas that cause collision will exhibit a much higher energy (see Fig.1) that automatically prevents the agent from walking through. No run-time collision detection is needed, as RRT does.  This is an obvious advantage since detecting collisions is often the most time consuming part.
> > > Also, we did evaluate the collision, as shown in Table 4, this demonstrates how accurately the paths model human behaviors.
> > >
> > > > **Q:**  A similar argument goes for the RRT being used with just 500 steps. Yes the NN can be much faster, and so having a limit on the number of steps is sensible, however, it would have been good to showcase the amount of time required by an RRT to find the optimum. This would not reduce the value of the proposed method but simply demonstrate the difference and tradeoff available.
> > >
> > > **A:** Please see the second row in Table 3.

---

> ### Author Response · Authors · 2021-11-22
> **For Reviewer z3Y4 (Part I)**
>
> Dear reviewer z3Y4, we thank you for the comments and respond to your comments below:
>
> > **Q:** *The issues start with it being unclear whether or not the idea is to reproduce something akin to a Dijkstra algorithm, i.e., any point to any point, or an A\* algorithm, i.e., any point to a single target location.*
>
> **A:** We respectfully disagree with the reviewer’s comments. As we stated in the first sentence in the abstract, this paper is about learning **“a novel scene representation that encodes reaching distance”**, not reproducing a  search algorithm. We have also explicitly discussed how our method, using a neural implicit function, is different from classical search algorithms in the whole third paragraph in our introduction. The two main differences are: a) **Continuity** and b) **Efficiency**. We wonder if the reviewer has missed the whole paragraph and we urge the reviewer to re-visit and look closer at our introduction to clear this misunderstanding.
>
> > **Q:** *The related work also lacks various aspects, such as voxelgrid and tsdf representations in the scene representation section or potential field methods and rmps in the path planning aspect.*
>
> **A:** Thanks for pointing out the related literature. We have revised our related work and cited the potential field literature in the “Path planning” paragraph. While we agree these all belong to scene representations, our work is mostly related to **neural implicit function**. This representation generally learns a neural network which takes a coordinate as inputs and outputs a signed distance value or occupancy. Please see the “Implicit neural representations.” paragraph in our related work for the most relevant literature.
>
> > **Q:** *The affordance prediction section also seems to be a misnomer as it focuses on trajectory prediction, which is different from affordances.*
>
> **A:** We respectfully disagree with the reviewer on this comment. What is affordance? When J.J. Gibson first introduced this concept, he defined affordance as opportunities of interaction in the scene. Going through the years, affordance has been used to refer to many different representations including segmentation masks [1], Interaction Hotspots [2], human pose [3], and human trajectories [4]. Thus the concept of affordance is actually very broad, and since our trajectory is interacting with the scene, it fits Gibson’s definition of affordance prediction.
>
> [1] Chuang et al. Learning to act properly: Predicting and explaining affordances from images. CVPR’18.
>
> [2] Nagarajan et al. Grounded human-object interaction hotspots from video. CVPR’19.
>
> [3] Wang et al. Binge Watching: Scaling Affordance Learning from Sitcoms. CVPR’17.
>
> [4] Wang et al. Synthesizing Long-Term 3D Human Motion and Interaction in 3D Scenes. CVPR’21.
>
> > **Q:** *The description of the method itself is simply devoid of all required detail, such as architecture, training regime, cost function regularization, biases induced due to the discretized nature of the training data, etc.”*
>
> **A:** We are not sure why the reviewer has the impression that there are no technical details in this paper, because technical details are all this paper is about. The other two reviewers do not seem to have this problem. In addition, Reviewer fAEP even specifically acknowledged **“The paper is well-written and is mostly easy to follow.”** as one of the strengths of this paper. We would appreciate the reviewer specifically **points out what is not right will be much more informative than saying everything is wrong**. We would appreciate more objective comments rather than one unclear sentence to dismiss months of research efforts.
>
> Since we do not know exactly what the reviewer is confused about, we point out all the implementation details written in the paper here. We are confident that the results can be reproduced by following our implementations:
> - Architecture: Section 3.1, 3.2, 3.4, second paragraph in Section 4.2, Fig. 3 and Fig. 5.
> - Training: Section 5.1 in the originally submitted paper and Section C.4 in the Appendix.
> - Cost functions: Section 3.2, especially equation (2). Second paragraph in Section 4.2.
>
> > **Q:** *The section that describes the human motion prediction… Furthermore, the general idea being described would likely fall into a sort of motion planning approach rather than a motion prediction method.*
>
> **A:** We agree that our work is related to motion planning. However, we distinguish it as human motion prediction since we not only require the human agent to not violate geometric constraints when moving, but also require the human to act naturally and perform behaviors such as sitting considering scene affordance. We believe this requires more knowledge compared to traditional motion planning. We will clarify this in our paper.

---

> ### Author Response · Authors · 2021-11-26
> **Please let us know if you have further questions**
>
> Dear Reviewer z3Y4,
>
> Thank you for your detailed comments. We have provided response based on your comments, please go over our response and let us know if you have further questions.
>
> Thanks!

---

### Official Review · Reviewer_fAEP · 2021-11-02

**Correctness:** 4
**Technical Novelty And Significance:** 3
**Empirical Novelty And Significance:** 3
**Recommendation:** 8
**Confidence:** 2

**Main Review:**

**Strengths:**
- Writing. The paper is well-written and is mostly easy to follow.
- Method / motivation. Proposed method seems novel and technically sound. Generally, the framework of implicit neural representations seems like a great fit for the task at hand.
- Experimental evaluation is conducted on several challenging tasks, a comprehensive comparison to baselines, including recent ones, is provided. Quantitatively, the method seems to be performing similarly to baselines or slightly better, while being more efficient.

**Weaknesses:**
- Reproducibility. There is not much details provided on the architectures of the networks. Figure 3 could benefit from more detailed explanations.
- Method. Authors claim that continuity is one of the core benefits of their approach, yet in practice they seem to be resorting to a discretized solution when doing trajectory search. Is there any benefit of using continuous representation in 2D? Say, over predicting a discretized field with a convnet.
- Experimental evaluation. It would be great to also have an idea about how the results look qualitatively for the baselines - it is hard to judge if the quantitative improvements are significant.
- Failure cases. There are some of the failure cases presented for the human motion prediction, but not for maze navigation. In particular, I wonder if the greedy strategy ever causes issues when the agent is stuck in an infinite loop between two locations, e.g. when the reaching distance is identical in those locations, and how one can tackle that?

**Misc:**
- I wonder if there is any analogy that can be made between the gradient-based navigation algorithm (Section 3.3) and a classical sphere tracing algorithm for rendering implicit surfaces.
- Searching-based -> search based?
- Supervisely -> in a supervised matter?
- a continues field -> continuous


**Summary Of The Paper:**

This work introduces a novel scene representation for agent navigation in 2D and 3D environments. At the core of the method is an implicit neural representation of environment - implicit environment field (IEF) - which is a neural net that maps location coordinates to its reaching distance. Several conditional variants of IEFs are proposed to allow for generalization to novel environments and goal locations. Navigation can the be achieved by gradient descent on the reaching function or a discretized greedy algorithm.
Experimental evaluation is conducted on 2D maze navigation and 3D human motion prediction, and indicate that the proposed representation performs favorably to the baselines.

**Summary Of The Review:**

This is a well-written paper that provides an interesting solution to an important problem. I am not too familiar with the field of motion planning, thus a low confidence in the assessment. From a more general perspective, the work seems technically sound, and the experimental evaluation, including baselines seem reasonable. With all that in mind, I am quite positive about this work, and recommend accept.

---

> ### Author Response · Authors · 2021-11-22
> **For Reviewer fAEP**
>
> Dear reviewer fAEP, Thanks for your constructive suggestions, we provide response to your comments below:
>
> > **Q:** *Reproducibility. There is not much details provided on the architectures of the networks. Figure 3 could benefit from more detailed explanations.*
>
> **A:** For Fig.3 a) and b), we use a 5-layer MLP with Sine activation as in [1]. For Fig.3 c),  the structure of the image encoder is: Conv -> ReLU -> 6 Residual blocks -> Conv, followed by the 5-layer MLP with Sine activation. The structure of the Residual block is Conv -> ReLU -> Conv -> ReLU.
> [1] Vincent Sitzmann et al. "Implicit neural representations with periodic activation functions." NeurIPS, 2020.
>
> > **Q:** *Method. Authors claim that continuity is one of the core benefits of their approach, yet in practice they seem to be resorting to a discretized solution when doing trajectory search. Is there any benefit of using continuous representation in 2D? Say, over predicting a discretized field with a convnet.*
>
> **A:** Please see the General Response above.
>
> > **Q:** *Experimental evaluation. It would be great to also have an idea about how the results look qualitatively for the baselines - it is hard to judge if the quantitative improvements are significant.*
>
> **A:** For the trajectories predicted by RRT and PRM, we have visualized their searching process in Fig. 10 of the Appendix of the originally submitted paper. We include visualizations of the HMP method in Fig.14 in the Appendix of the revised manuscript.
>
> > **Q:** *Failure cases. There are some of the failure cases presented for the human motion prediction, but not for maze navigation. In particular, I wonder if the greedy strategy ever causes issues when the agent is stuck in an infinite loop between two locations, e.g. when the reaching distance is identical in those locations, and how one can tackle that?*
>
> **A:** Due to equal reaching distance (i.e., a flat surface region), the agent may be stuck at a location during the trajectory searching process. Our solution is to sample from all possible locations in the next step, instead of directly taking the one with the smallest reaching distance. Specifically, for all 8 possible directions, we compute a probability vector formed as the inverse of their reaching distance. We treat this probability vector as a multinomial probability distribution and sample one direction out of 8. We have included the above details in Section Appendix C.2 of the revised manuscript.
>
> > **Q:** *I wonder if there is any analogy that can be made between the gradient-based navigation algorithm (Section 3.3) and a classical sphere tracing algorithm for rendering implicit surfaces.*
>
> **A:** Our method shares similar high-level intuition with the sphere tracing algorithm [1]. In both cases, one marches along a trajectory/ray towards a given goal (i.e., the intersection of the ray and the surface in ray tracing and the goal position in our case) based on the distance towards the goal. However, there are significant differences:
> - In sphere ray tracing, the step length (i.e., sphere radius) is adaptively determined by the distance implicit functions, whereas in our case, the step length is determined by the given human motion sequence.
> - In sphere ray tracing, the direction is determined by the ray and is the same for every step. However, in our case, the direction is determined at each step based on the distances of each possible next location to the goal and is constantly changing.
>
> [1] John C. Hart  "Sphere tracing: A geometric method for the antialiased ray tracing of implicit surfaces." The Visual Computer 12.10 (1996): 527-545.
>
> > **Q:** *Typos*
>
> **A:** Thanks for pointing out the typos, we have fixed them and marked them red in the revision.

---

> > ### Comment · Reviewer_fAEP · 2021-11-24
> > **Re:**
> >
> > Thanks for an elaborate response. I am keeping my original rating.

---

### Official Review · Reviewer_AhgQ · 2021-11-02

**Correctness:** 3
**Technical Novelty And Significance:** 2
**Empirical Novelty And Significance:** 2
**Recommendation:** 6
**Confidence:** 4

**Main Review:**

## Strengths
- I believe the main strength of the proposed solution is the ability of the learned value function to generalize to different goals & different scenes, which is ensured by conditioning the network on a target goal and learned features from the 2D layout of the specific scene in question (e.g. section 3.4 and Fig. 3, c)). It is nice to see that an MLP can indeed generalize successfully when different goals & scene layouts are fed in, as illustrated by the multiple human navigation experiment (shown in Fig. 4). Section 3 also mentions predictions are made for unseen mazes during inference, which is good.
- The showcased value-function field evaluations in Fig. 1 and Fig. 6 also appear meaningful, capturing the correct geodesics in the scenes and showing useful predictions when the goal is switched.
- Planning based on the learned representation seems possible, with good success rates for the considered environments, albeit with the assumed discretization of 8 (in 2D) or 27 (in 3D) directions of movement mentioned in sections 3.3 and 4.2 (see my point below).

## Weaknesses
- Looking at figure 2, c), which shows an example of obtaining a navigation plan via gradient descent through the learned representation, it seems like the trajectory is inconsistent and unstable at times. A main advantage of the proposed value function representation is that it is continuous (compared to the discrete data it is trained on), which would require meaningful interpolation in the regions between training data. I am worried that, given the example in figure 2, the gradients (and respectively predictions) of the network in these interpolation regions (i.e. between the points from FMM) might be suboptimal. Can you please comment on this? Right now I am interpreting this as signs of overfitting.
- Along the same line of reasoning as the previous point, constructing plans from the learned network by choosing from discrete directions of movement (8 for 2D, 27 for 3D) seems inconsistent with the continuity argument. To me it seems like a step backwards towards maintaining a discrete tabular value function. Ideally, gradient descent should work.
- The idea of learning conditional value functions for goal-reaching tasks that generalize to different goals is not exactly novel, e.g. works like [2] have already considered generic formulations in that regime that go beyond holonomic navigation in $\mathbb{R}^2$ or $\mathbb{R}^3$ (you can check the references section of the mentioned paper for more examples of works from the "goal-reaching RL" field). More generally, learning continuous value functions is standard practice in the RL and optimal control communities, and I think the link to that should be more emphasized in the paper.
- When modeling accessible regions in 3D space with a VAE (section 4.2), what is the motivation behind using the VAE model to begin with, i.e. why not directly use the empirical distribution of human torso locations the VAE is trained on? I am asking because the trained generative part of the VAE will be a proxy for the empirical distribution it was trained to approximate. In section 4.2 there is also the note that samples from the generative model are further filtered out by checking for collisions–this makes it hard to judge if the generative model is indeed reflective of accessible regions, or if this resampling / filtering step is mostly responsible.
- Section 4.2 mentions locations outside of the accessible region are marked with negative value, but how do you determine whether a point lies in the region or not? When using the generative part of the VAE (or the data empirical I mentioned, for that matter), this region would not be defined in a closed form. This needs further clarification.
- NERF [3] should probably be cited in the related work section on implicit neural representations.

## Other remarks
- Looking at the supplementary material, there are some accidental flips of the human skeletons in the videos. Is this because of issues with the alignment of the movement primitives to the planned trajectories, or because the planned trajectories have occasional jitter?


## References
[1] James A Sethian. A fast marching level set method for monotonically advancing fronts. PNAS, 1996

[2] Eysenbach, Benjamin, Ruslan Salakhutdinov, and Sergey Levine.  "C-learning: Learning to achieve goals via recursive classification." ICLR 2020

[3] Mildenhall, Ben, et al. "Nerf: Representing scenes as neural radiance fields for view synthesis." ECCV 2020.



**Summary Of The Paper:**

The paper proposes modeling reaching distance between any start position and any goal (subject to obstacle avoidance) with a neural network. This is equivalent to parameterizing a traditional path-planning (goal-reaching) continuous value function with the network, which the authors also mention in the introduction section. Different variants of the network are considered, with conditioning on goal (aiming to generalize to different goals) and conditioning on a 2D scene layout (aiming to generalize across different scenes). The network is trained in a supervised manner on data obtained from a traditional search method (fast marching method [1]) that assumes discrete states. The usefulness of the trained value network for navigation and its generalization properties are then experimentally validated in 2D and 3D environments, including an interesting navigation example with two dynamically moving agents in the same scene.

**Summary Of The Review:**

Because of the empirically validated generalization properties of the presented model when the 2D scene layout & goal inputs are switched, I am leaning slightly positive, but currently the concerns listed in my review are stopping me from giving a higher score.

---

> ### Author Response · Authors · 2021-11-22
> **For Reviewer AhgQ**
>
> Dear reviewer AhgQ, Thanks for the detailed comments and suggestions, we provide response to your comments below:
>
> > **Q:** *Continuity of the environment field.*
>
> **A:** Please see the General Response above.
>
> > **Q:** *NERF [3] should probably be cited in the related work section on implicit neural representations.*
>
> **A:** NERF is a special form of implicit functions, and we have cited it in the revised manuscript.
>
> > **Q:** *The idea of learning conditional value functions for goal-reaching tasks that generalize to different goals is not exactly novel, … More generally, learning continuous value functions is standard practice in the RL and optimal control communities, and I think the link to that should be more emphasized in the paper.*
>
> **A:** We have differentiated our model and RL methods in the “Path planing” paragraph of Section 2.
>
> > **Q:** *When modeling accessible regions in 3D space with a VAE (section 4.2), what is the motivation behind using the VAE model to begin with, i.e. why not directly use the empirical distribution of human torso locations the VAE is trained on?*
>
> **A:** We use the VAE because:
> - As discussed in Section 4.2, the “accessible regions” in a 3D room refer to places that humans commonly appear. The human trajectory data the VAE trained on is sparse and does not cover the entire 3D room. For instance, there are only about two human trajectories for each 3D room in the PROX dataset. Thus, the training data cannot be directly used to learn a dense environment field.
> - Instead, we use a VAE to learn human behavior statistics and generate more locations for humans to plausibly appear in the room. For example, if a human in the training dataset sits on a sofa, then the VAE should learn that other regions similar to the sitting spot on the sofa are also suitable for humans to sit.
> - Thus, to train the environment field, we use the VAE to sample more locations that are plausible for humans to appear. As shown in Fig.7 (b), the “accessible region” generated by the VAE is much more dense than locations covered by few human trajectories.
>
> > **Q:** *In section 4.2 there is also the note that samples from the generative model are further filtered out by checking for collisions–this makes it hard to judge if the generative model is indeed reflective of accessible regions, or if this resampling / filtering step is mostly responsible.*
>
> **A:** The VAE does indeed capture human behavior patterns for two reasons:
> - It can discover “sitting spots” for humans. For instance, in Fig.7 (b), there are some “suitable” locations for humans on the sofa.
> - The VAE learns to predict accurate walking human torso locations. The filtering process only adopted for eliminating locations where the bird’s eye view projections fall onto walls in order to avoid potential collisions, it *cannot* fix inaccurate tosor locations, e.g., with wrong height, where the VAE is mainly responsible for.
>
> > **Q:** *Section 4.2 mentions locations outside of the accessible region are marked with negative value, but how do you determine whether a point lies in the region or not?”*
>
> **A:** We sample a large number of possible locations from the VAE by feeding noise into the VAE, we also include the points from the training data. Then when computing the ground truth reaching distance by the FMM method, we discretize the 3D scene into a voxel grid, any grid that is occupied by one sampled grid is considered as positive (i.e., within accessible region), other grids are considered negative (i.e., outside the accessible region)
>
> > **Q:** *Looking at the supplementary material, there are some accidental flips of the human skeletons in the videos. Is this because of issues with the alignment of the movement primitives to the planned trajectories, or because the planned trajectories have occasional jitter?*
>
> **A:** The flip is caused by the alignment process. As discussed in Appendix D, the randomly sampled pose sequence may not well fit the predicted trajectory. Thus the motion of the aligned human pose sequence includes flickering. One future work is to generate human poses besides trajectory given the environment field such that the flickering can be reduced by the pose prediction model.

---

> > ### Comment · Reviewer_AhgQ · 2021-11-28
> > **Reply to authors**
> >
> > Thank you for addressing the points listed in my review.
> >
> > The response provides clarity about the usage of the VAE model & the nature of the observed flips in the supplementary videos, which is appreciated. I hope the details & motivation listed here will also be included in the appendix of the final version of the paper.
> >
> > As noted in the first three points in my review, my main reservations were about the continuity of the learned value function & establishing a strong connection to existing prior-work in optimal control & RL. In terms of the former: I understand that SGD can overshoot, but then the question still stands whether the issue is in the step size or in the interpolation (overfitting). This point is important, as the continuity of the model is one major aspect that distinguishes it from tabular representations (or simple interpolation schemes thereof, like trilinear interpolation in 3D). As for positioning the work: value functions like the one presented in this work are a traditional concept in optimal control & reinforcement learning, e.g. as discussed in [1], and learning continuous critics is very common (e.g. actor-critic methods). As noted in my original review, this link should be in the forefront to make the contributions of the work precise, but I understand that this might require a substantial revision of the introduction section.
> >
> > With all this in mind, I believe my original score still applies, leaning slightly positive primarily due to the shown generalization properties (when the conditioning on start & goal changes).
> >
> > [1] Bertsekas, Dimitri. Dynamic programming and optimal control: Volume I. Vol. 1. Athena scientific, 2012.

---

> > > ### Author Response · Authors · 2021-11-30
> > > **Further Clarification**
> > >
> > > Dear reviewer AhgQ,
> > >
> > > Thanks for your response. We would like to further respond to your concerns as follows:
> > >
> > > > **Q:** *As noted in the first three points in my review, my main reservations were about the continuity of the learned value function & establishing a strong connection to existing prior-work in optimal control & RL. In terms of the former: I understand that SGD can overshoot, but then the question still stands whether the issue is in the step size or in the interpolation (overfitting). This point is important, as the continuity of the model is one major aspect that distinguishes it from tabular representations (or simple interpolation schemes thereof, like trilinear interpolation in 3D).*
> > >
> > > **A:** The issue is in the step size rather than overfitting. First, we do not see a divergence of performance between the training and the testing sets. Second, we observe that learned fields are smooth, e.g., in Fig. 6 (b) and the supplementary video. The smoothness of the field can also be guaranteed via the continuity of the implicit function, i.e., the output values of nearby locations will also be close to each other. We will emphasize this point in the revised paper.
> > >
> > > > **Q:** As for positioning the work: value functions like the one presented in this work are a traditional concept in optimal control & reinforcement learning, e.g. as discussed in [1], and learning continuous critiques is very common (e.g. actor-critic methods). As noted in my original review, this link should be in the forefront to make the contributions of the work precise, but I understand that this might require a substantial revision of the introduction section.
> > >
> > > **A:** While we acknowledge that the environment field can be considered as a continuous value function (see related work), our key contribution is to model it via an implicit function. Thus, the model enjoys many advantages of implicit function, e.g., a). It is efficient, since the values for all locations can be predicted by a single network pass; b) As a neural representation, this model is easy to extend, e.g., to model human behavior jointly, which goes beyond the navigation task. We will add this clarification to the next revision.

---

> ### Author Response · Authors · 2021-11-26
> **Please let us know if you have further questions**
>
> Dear Reviewer AhgQ,
>
> Thank you for your detailed comments. We have provided response based on your comments, please go over our response and let us know if you have further questions.
>
> Thanks!

---

> ### Author Response · Authors · 2021-11-28
> **Please let us know if you have further questions**
>
> Dear Reviewer AhgQ,
>
> We appreciate your constructive comments and have responded to address your concerns. As the discussion period will end soon, please review our response and let us know if you have additional questions.
>
> Thank you!

---

### Author Response · Authors · 2021-11-22
**General Response and Revision Summary**

We thank all reviewers for their constructive comments. We first answer the common questions from the reviewers and summarize changes in the revision below. All changes in the revision are marked in red.

**Continuity of the environment field**
- **(To all):** Our environment field is represented as an implicit function, which inherently learns a continuous surface given discretely sampled training points [1].
- **(To all):** The key advantage of a continuous environment field is that it can output the “reaching distance” at any location in a scene. We indeed utilize such property in the bird’s eye view trajectory search (Section 4.1) and “Pose-dependent trajectory from a continuous field” (Section 4.2). Taking the trajectory search in the 3D room as an example, at each step we are given the step length of a human and aim to choose one out of 27 directions to take. We emphasize that the step length depends on the given pose sequence. Thus the human can appear at any location in the scene. But thanks to the continuity of the environment field, we can predict the reaching distance for any location in the scene and plan for the next step for the human accordingly. We note that here we quantize the directions to 27 choices since otherwise the possible locations for the next step are infinite -- they form a sphere that is centered at the current location with the step length as its radius. It is not possible to search in a space with infinite directions.
- **(To AhgQ):** Naively using gradient descent (as in Fig.2, c) for trajectory search is unstable, for two reasons: (i) The step-length of SGD is determined empirically. It is known to produce unstable trajectories, e.g., overshooting, even for a simple convex function, e.g., a ​​parabola. (ii) While the learned environment field is continuous, we adapt the discrete searching strategy, in order to follow the standard experimental setting for 2D maze (i.e., an agent is constrained to take one grid at one of eight directions in each step), and make a fair comparison with the other baselines.
We also note that the 2D continuous field can be used beyond the simple 2D maze scenario. For instance, it is used in Sec. 4.1, where a continuous field is essentially needed for varying step-length of the humans in the scenes.

[1] Jeong Joon Park et al. "Deepsdf: Learning continuous signed distance functions for shape representation." CVPR. 2019.

**Reproducibility**
- **(To fAEP and z3Y4)** Code will be released upon publication as mentioned in the reproducibility statement.

**Summary of changes in revision**
- Typos are fixed. (fAEP)
- Details to solve failure cases in 2D maze trajectory search are added to Appendix C.2. (fAEP)
- Qualitative comparison with baseline method (HMP) is added to Appendix D.4. (fAEP)
- Citation of Potential field is added to page 3. (z3Y4)

---

### Decision · Program_Chairs · 2022-01-20

**Decision:**

Accept (Poster)

**Comment:**

This paper proposes environment fields, a representation that models reaching distances within a scene. Dense environment fields are learnt using a neural network, and the effectiveness of this representation is shown on 2D maze environments and 3D indoor environments. This paper received hugely contrasting reviews, with two reviewers being very supportive and one reviewer providing the lowest score of 1. In light of this, I'll start with providing my takeaways on the review and discussion with reviewer z3Y4 (rating of 1) and then proceed to the remaining discussion.

Reviewer z3Y4 has provided the score of 1 and has made strong remarks that include: "what is proposed in this paper is simply not comprehensible", "description of the method itself is simply devoid of all required detail", "The main claims of the paper are incorrect or not at all supported by theory or empirical results." and " what is being proposed in this paper is simply too unclear and vague to be assessed". **Such dismissive remarks, in my opinion, are completely unnecessary and create a toxic discussion and review environment.**

Reviewer z3Y4 has many criticisms of the submission, but the primary ones include: (a) the lack of details throughout the paper (b) the positioning of the paper in the abstract and introduction, and (c) the lack of experiments in continuous environments. Re (a): It is well understood in our research community that providing every last detail in the main submission is nearly impossible due to the restriction on the number of pages. Providing excess details in the main paper also often reduces the readability of the paper. Such details are better addressed in the appendix and crucially, the code. The authors have provided some details in the appendix and have indicated that they will release a code base.  I also agree with the authors that justifying every last detail in the network architecture such as choice of an activation function is not necessary for this submission. The same goes with describing methods in past works in detail vs referring the reader to the appropriate citation. As a result, I believe that the authors have addressed (a) well. Re (b): This has also been addressed by the authors, by pointing out relevant parts of the paper that had the necessary details. Re (c): In this regard, the paper clearly contains a well laid out experiment in 3D indoor scenes, so as far as I am concerned, this has been addressed in the main submission.

Reviewers AhgQ and fAEP have supported this submission but also laid out some concerns that include:
(1) Are the gradients suboptimal ?
(2) Positioning the paper with regards to past works
(3) Motivation behind using the VAE
(4) Qualitative analysis and failures
The authors have addressed these 4 concerns well using the rebuttal as well as via a revision of the appendix. The reviewers, post discussion have indicated their satisfaction with the revised submission.

I think this paper is interesting and proposes a novel scene representation which can be useful for others in the Embodied AI community. I am in agreement with reviewers AhgQ and fAEP, and in spite of the strong reject score by z3Y4, I recommend accepting this paper.